# Programming Effect of the Parental Obesity on the Skeletal System of Offspring at Weaning Day

**DOI:** 10.3390/ani11020424

**Published:** 2021-02-06

**Authors:** Radoslaw Piotr Radzki, Marek Bienko, Dariusz Wolski, Monika Ostapiuk, Pawel Polak, Malgorzata Manastyrska, Aleksandra Kimicka, Joanna Wolska

**Affiliations:** 1Department of Animal Physiology, Faculty of Veterinary Medicine, University of Life Sciences in Lublin, Akademicka 12, 20-033 Lublin, Poland; darek.wolski@o2.pl (D.W.); manastyrska.m@gmail.com (M.M.); aleksandra.kimicka@gmail.com (A.K.); 2Department of Materials Engineering, Lublin University of Technology, Nadbystrzycka 36, 20-618 Lublin, Poland; m.ostapiuk@pollub.pl; 3St Johns’ Oncology Center in Lublin (COZL) Trauma, Orthopaedic Surgery Department, ul. Jaczewskiego 7, 20-090 Lublin, Poland; ppolak.ort@gmail.com; 4Department of Oncology, Chair of Oncology and Environmental Health, Faculty of Health Sciences, Medical University of Lublin, 20-090 Lublin, Poland; wolska.joanna@o2.pl

**Keywords:** prenatal programming, nutrition, bone metabolism and development, metabolic diseases, osteoporosis, bone mechanical examination, densitometry, pQCT, µCT

## Abstract

**Simple Summary:**

Overweight and obesity can cause many diseases, and several studies indicate a close relationship between the obesity of parents and the health of their offspring. Our aim was to investigate whether there is a programming influence of parental obesity on the skeletal system in weaned female and male offspring rats. In undertaking this, analysis of bone material was carried out using isolated tibia, and densitometry (DXA), peripheral (pQCT) and micro (µCT) computed tomography were performed. Mechanical tests and blood serum biochemistry were also carried out. Our work showed a significant programming influence of parental obesity on neonatal skeletal development. The tibiae isolated from offspring originating from obese parents were characterized by more intense mineralization and higher fracture resistance. However, numerous studies demonstrate the destructive effect of obesity on the skeletal system. Our research and the available literature suggest the existence of a “fat threshold”, the exceeding of which changes of the osteotropic effect of adipose tissue to become unfavorable. Therefore, there is a need for further research to determine the time-dependent metabolic relationship between adipose tissue and bone in both animals and humans.

**Abstract:**

Our study aimed to verify the hypothesis of the existence of a programming effect of parental obesity on the growth, development and mineralization of the skeletal system in female and male rat offspring on the day of weaning. The study began with the induction of obesity in female and male rats of the parental generation, using a high-energy diet (group F). Females and males of the control group received the standard diet (group S). After 90 days of dietary-induced obesity, the diet in group F was changed into the standard. Rats from groups F and S were mated to obtain offspring which stayed with their mothers until 21 days of age. Tibia was tested using dual-energy X-ray absorptiometry (DXA), peripheral quantitative computed tomography (pQCT), micro-computed tomography (µCT) and mechanical strength using the three-point bending test. Biochemical analysis of blood serum bone metabolism markers was performed. DXA analysis showed higher tibia bone mineral content (BMC) and area. pQCT measurements of cortical and trabecular tissue documented the increase of the volumetric bone mineral density and BMC of both bone compartments in offspring from the F group, while µCT of the trabecular tissue showed an increase in trabecular thickness and a decrease of its separation. Parental obesity, hence, exerts a programming influence on the development of the skeletal system of the offspring on the day of the weaning, which was reflected in the intensification of mineralization and increased bone strength.

## 1. Introduction

Nowadays, the problem of overweight and obesity has a global dimension and concerns both animals and humans. The main reasons for the development of this pathology are a diet rich in carbohydrates and fats, as well as low physical activity. Recently, much attention has been focused on the programming impact of parental obesity on the growth and development of offspring [1,2]. Some studies show that parental obesity may significantly contribute to the development of obesity of offspring, and as a consequence, of metabolic syndrome [3,4]. What is more, accordingly, obesity exerts a programming influence on the development of various organs [5].

Adipose tissue is the main energy supply in the body, but is also one of the largest endocrine organs, synthesizing and releasing several adipokines that affect the entire body, including bone [6]. The effect of adipose tissue is expressed not only through the direct effect of adipokinins on bone tissue, but also through their indirect, regulatory effect on the activity of other hormones important for metabolism, such as growth hormone (GH) or 1.25(OH)2D3 [7,8]. The osteotropic effect of adipose tissue is multidirectional and its nature has not been clearly defined. It is suggested that an increase in body weight is accompanied by stimulation of bone tissue formation by an increase in osteoblast proliferation and differentiation, reducing their apoptosis [9,10]. In contrast, excessive weight loss and reduced pressure on the pelvic limb bone, or its immobilization, intensify bone resorption [11]. There are reports indicating the different effect of excessive fat accumulation on bone tissue. Blum et al. [12] and Hsu et al. [13], for example, show that excessive fat mass is not a protective factor against metabolic diseases of bone tissue and even contributes to their development, which is low with low BMD and BMC.

Skeletal development during the perinatal period and in the subsequent stages of life before adolescence is important in achieving peak bone mass. It is estimated that about 30% of the peak bone mass depends on nutrition [14,15,16]. It should be emphasized that the composition of milk of obese mothers significantly differs from the composition of milk of mothers with normal body weight [17]. Such modification of nutrition in the perinatal period may have a significant influence on the development of bone tissue.

The study focused on determining whether and to what extent the established dietary-induced obesity in parents before fertilization exerts a programming effect on the development of the female and male offspring’s skeletal system on the weaning day. This research could significantly contribute to a better understanding of the relationship between obesity and bone tissue.

## 2. Materials and Methods

### 2.1. Animal Procedures

All the procedures were approved by the 2nd Local Ethics Committee in Lublin (Resolution nr 32/2015). The experimental population consisted of 32 female and 16 male Wistar rats (outbred herd from the Center of Experimental Medical in Bialystok Medical University–Cmdb:Wi, originating from Charles River) with initial body weight 230 ± 20 g. These rats were the parental generation. Throughout the experiment, the animals were exposed to artificial lighting, with the day/night ratio being 12/12 h, except during the breeding period in which case it was 14/10 h. Access to water and food was not restricted, and body weight was monitored once a week. The rats were randomly divided into two groups and fed with diets of different energy values. Group S (females *n* = 16, males *n* = 8) received a standard diet with an energy value of 11.5 MJ/kg, while group F (females *n* = 16, males *n* = 8) received a high-energy diet valued at 17.6 MJ/kg (16% fat, 65% carbohydrates, 19% protein) (Agropol, Motycz, Poland). Both types of diets were followed for 90 days, i.e., the time necessary to induce obesity in groups of rats receiving diet F [18,19] (Table 1) and then a high-energy diet in group F was changed into a standard diet. After 90 days the mean body weight of females in the S group was 383 g ± 9, and in the F group 484 g ± 14 (*p* < 0.0001), whereas in the males of the S and F groups, the mean body weight was 632 g ± 18 and 766 ± 18 (*p* < 0.0001), respectively.

In the next step of the experiment female and male rats of the parental generation were placed in shared cages (2 females and 1 male in a cage) for 14 days (the time when fertilization should occur) to obtain offspring. The parental females were then separated and housed in individual cages. The average litter did not exceed 8–12 newborns. On the first day after birth, each litter was examined and initially sexed based on the method previously described [20], and the weight of all newborns was measured. Six offspring from the S and the F parents were selected and left with their mother for the last step of the study (16 females × 6 litters in each feeding group). The selection criterion was body weight, being close (max ± 0.5 g) to the mean body weight of all females or males offspring in either the S or the F group, respectively. On the seventh day after birth, the sex was confirmed and the body weight was measured again. Measurements were repeated at day 14 and 21. On the weaning day (21st day of life), 16 female offspring (group S *n* = 8; group F *n* = 8) and 16 male offspring (group S *n* = 8; group F *n* = 8) were randomly selected and sacrificed. Immediately after sacrifice, subcutaneous, visceral, mesenteric fat tissues, as well as tibiae, were dissected and weighted. The length of the tibiae was established via pQCT analysis. The isolated tibia as well as blood serum were frozen (−30 °C) for further analysis.

### 2.2. Dual X-ray Absorptiometry (DXA) Analysis of the Total Skeleton and Isolated Tibiae

Lean mass (LM), fat mass (FM), areal total skeleton bone mineral density (Ts.BMD), total skeleton bone mineral content (Ts.BMC) and total skeleton area (Ts.Ar), as well as t.BMD; t.BMC and t.Ar for isolated right tibiae were examined via DXA analysis. The measurements were performed with the use of a Norland Excell Plus Densitometer (Fort Atkinson, WI, USA) equipped with Illuminatus Small Subject Scan v.4.5 software.

### 2.3. Peripheral Quantitative Computed Tomography (pQCT) Measurements of the Isolated Tibiae

The right tibiae were measured with the use of the pQCT XCT Research SA Plus system and software version 6.2 C (Stratec Medizintechnik GmbH, Pforzheim, Germany). The bones were scanned perpendicularly to the long axis. The scan location was established using the scout view obtained after the initial pQCT system scan (pre-scan). The pre-scan was performed at a speed of 10 mm/s and the CT scan was done at 4 mm/s. The cortical and trabecular bone tissue are separated by an appropriate superficial distribution of both bone compartments, and 55% of the outer surface of the bone is defined as the cortical-subcortical area and 45% of the internal core is defined as trabecular bone tissue. After the CT scan, the XCT Research algorithm based on the above settings (threshold, contour mode, peel mode, cortical mode) automatically calculated values for densitometric and architectural properties. Trabecular bone analyses were performed in contour mode 2 and a peel mode 20 (threshold 610 mg/cm^3^) in the proximal tibia metaphysis. The following parameters were obtained: total bone mineral content (Tot.BMC), total volumetric bone mineral density (Tot.vBMD), total surface bone area (Tot.Ar), trabecular bone surface area (Tb.Ar), trabecular bone mineral content (Tb.BMC) and trabecular volumetric bone mineral density (Tb.vBMD). In the analysis of cortical bone tissue in the mid-shaft of the tibia diaphysis, cortical mode 1 (threshold 710 mg/cm^3^) was used to obtain the following parameters: total and cortical volumetric bone mineral density (Tot.vBMD; Ct.vBMD), total and cortical bone mineral content (Tot.BMC; Ct.BMC) and cortical surface (Ct.Ar). Total area (Tot.Ar), periosteal (Peri. C) and endosteal (Endo C) circumferences and cortical thickness (Ct.Th) were measured using contour mode 1 (710 mg/cm^3^ threshold) to determine the outer edge of the bone, and peel mode 2 (400 mg/cm^3^ threshold) was used to separate the cortical and subcortical/medullary compartments. The Ct.Th, Endo.C and Peri.C measurements were based on a circular ring model where the spinal area was defined as the difference of total area and cortical area. Before measurements, a daily system calibration was performed using a hydroxyapatite standard, containing a quality control phantom (pQCT QA-Phantom).

### 2.4. 3D Micro-Computed Tomography (µ-CT) Analysis of Trabecular Bone Tissue Morphometry

The micro CT technique with the use of a Micro CT SkyScan 1174 (Bruker-SkyScan, Kontich, Belgium) was performed to examine the proximal part of the tibia. The micro CT apparatus was equipped with a VDS 1.3Mp FW camera (resolution of 1024 × 1024), a 0.5 mm aluminum filter, and a lamp of 50 kV and 800 µA. All tibia scans were performed 1.5 mm from the proximal growth plate, with a rotation degree of 0.70° and 2400 ms of time exposure. The mean number of frames was 4, image pixel size was 9.64 μm, and the average time of scanning was 1 h and 5 min 50 s. With the use of NRecon 1.6.1.5 software (SkyScan n.v., Kontich, Belgium) the reconstructions of the tibia were performed. Analysis of the internal structure was generated by CT Analyser software ver. 1.15.4.0 (SkyScan n.v., Kontich, Belgium) which allowed the obtaining of three-dimensional reconstructions and 3D information by stacking measured slices on top of each other. The following parameters were measured: bone volume (BV), total volume (TV), bone volume density (BV/TV), bone surface (BS), object surface density (BS/TV), trabecular number (Tb.N), trabecular thickness (Tb.Th), trabecular separation (Tb.Sp) and structure model index (SMI).

### 2.5. Mechanical Properties

The mechanical strength of bone tissue was determined by two methods. Tomographic analysis (pQCT) allowed for the establishment of the Strain/Strength Index (xSSI), enabling prediction of the mechanical strength of the bone. These measurements were carried out in the middle of the tibia’s diaphysis (the measurement conditions described above). The real mechanical parameters of the isolated tibia were established based on the 3-point bending test, using a ZwickRoell Z010 (ZwickRoell GmbH & Co. KG, Ulm, Germany) testing machine with a 1 kN measuring head Xforce HP series. The analyzed bones were regarded as a tube model. Herein, the external and internal diameters were measured by pQCT. In doing so, the bone was placed on two holders, and the force was applied downward, perpendicularly to the long bone axis, at the midshaft of the bone column. The received data were analyzed by way of testXpert II 3.1 software assistance, and the ultimate strength (Fmax), work to ultimate strength (W/Fmax), and the Young modulus (Emod) were determined [21].

### 2.6. Biochemical Analysis

The serum concentration of osteocalcin (OC), also known as bone gamma-carboxyglutamic acid-containing protein (BGLAP) (Immunodiagnostic Systems, Bolton, UK), C-terminal telopeptides of type I collagen (CTX-I) (Immunodiagnostic Systems, UK), 25 (OH)D3 (Immunodiagnostic Systems, UK) and bone-specific alkaline phosphatase (Immunodiagnostic Systems, UK) were assessed through enzyme-linked immunosorbent assay (ELISA), using the respective commercial kits. Ionized calcium and phosphorus were measured spectrophotometrically by way of commercial kits (Alphadiagnostic, Warsow, Poland) and a Mindray BS 120 apparatus (Mindray, Shenzhen, China).

### 2.7. Statistical Analysis 

The results were presented as mean values ± S.E.M (*n* = 8). To establish significant differences between control and experimental groups, the *t*-Student test was used. Differences were considered significant at *p* < 0.05. Statistical analyses were performed using STATISTICA software v. 13.0 (Tibco Software Inc., Palo Alto, USA). 

## 3. Results

### 3.1. Body Weight, Body Composition, Tibia Length and Mass

Body weight of newborns from parents fed the F diet, measured on the day of birth, was higher in both females (15%) and males (12%) compared to the mean birth weight of the rats from the parents fed the S. At weaning, these differences were 17% and 29%, respectively (Table 2). 

Fat Mass (FM) in both female and male offspring from group F was statistically significantly higher. Interestingly, LM and ST also tended to higher values in both sexes of offspring in group F. The weight and length of the tibia in both females and males in groups F and S were similar (Table 2).

### 3.2. Mass of the Subcutaneous, Visceral and Mesenteric Fat Tissue

The weight of the subcutaneous and mesenteric fat was significantly higher in female rats from group F (vs. S) by 21% and by 40%, respectively. Surprisingly, we noted an almost 2.6 times increase of the weight of visceral fat in females from group F. In males, a significant increase in the depot of subcutaneous, mesenteric and visceral fat tissue by 25%, 28% and about by 140% respectively, were also observed (Table 2).

### 3.3. DXA Measurements of the Whole Skeleton and the Isolated Tibia

There were no significant differences in the total skeleton Ts.BMD values between groups F and S. Diet F used in the parents, however, significantly increased the Ts.BMC (by 30% in both sexes) and the Ts.Ar in the female and male offspring (by 34% and 33% respectively). The tibia of both sexes of offspring from parents fed the F diet, showed a statistically significant increase in t.BMC (23% and 27%, respectively) and t.Ar (16% and 20%, respectively) with similar values of t.BMD (Figure 1). 

### 3.4. pQCT Analysis of Tibia 

The usage of high-energy diet F significantly increased the values of Ct.BMC and Ct.vBMD of the tibia in offspring females (9% and 7%, respectively) and males (12% and 7%, respectively). Tibia diaphysis of offspring in group F was also characterized by a significantly greater thickness of cortical bone tissue (Ct.Th) by 4% and by 9%, respectively. The values of the other parameters (Ct.Ar, Peri.C and Endo.C) of the tibia were similar between the sexes in groups S and F (Table 3).

Statistically significant increase of the value of volumetric mineral density (Tb.vBMD) (55% and 75%, respectively) and the content of trabecular bone mineral tissue (Tb.BMC) (43% and 45%, respectively) were noted in both females and males from group F. The trabecular bone area (Tb.Ar) of the proximal part of the tibia in groups S and F remained at a similar level (Table 3). 

### 3.5. µCT Analysis of Trabecular Bone Tissue in the Proximal Metaphysis

Micro CT measurements showed a significant increase in bone volume fraction (BV/TV) in females (25%) and males (37%) of group F, while in the S groups, object surface density (BS/TV) significantly decreased (11% in both sexes). The dietary-induced obesity of parents significantly decreased the Tb.N (20% and 17%, respectively), decreased the Tb.Sp (24% and 29%, respectively) and increased Tb.Th (43% and 45%, respectively) measurements in female and male offspring. Moreover, the structure model index (SMI) decreased in offspring from the F group by 76% in females and by 88% in males (Table 4).

### 3.6. Mechanical Analysis of Tibia

Tomographic examination showed that the strain/strength index (xSSI) of the tibia of the offspring of rats from group F was significantly higher by 14% in females and by 11% in males. The evaluation of the mechanical properties by the three-point test of the tibia from females and males of group F also showed a significant increase in maximum strength (Fmax) (63% and 89%, respectively). Besides, females and males of group F revealed Young’s modulus of elasticity (Emod) of 60% and 60%, respectively, and the work to ultimate strength (W/Fmax) of 20% and 53%, respectively (Figure 2).

### 3.7. Biochemistry 

Osteocalcin concentration significantly increased in the blood serum of females (7%) and males (8%) from group F, while the CTX-I significantly decreased in both sexes (14% and 18%, respectively). The activity of the ALP in offspring from the parents fed the high-energy diet significantly increased in both sexes. The level of 25(OH)D3 concentration in blood serum of females offspring from group F (6%) tended to lower values as compared to the S group, while in males (7%), a significant increase of the 25(OH)D3 concentration in the F group was noted. No significant changes were evidenced in the concentration of ionized Ca and P (Table 5).

## 4. Discussion

Our research was conducted to determine the programming influence of persistent parental obesity on the mineralization, structure and mechanical strength of bone tissue of rat offspring on the day of weaning. Offspring of both sexes came from parents with fixed dietary-induced obesity by high-energy diet (F) and from parents fed the standard diet (S). We used a tibia as a model bone. 

DXA is now the gold standard in the study of the skeletal system of humans and animals. The test is readily available, the results are highly comparable and the patients are exposed to a low dose of radiation. An additional benefit of the DXA is the ability to evaluate the Body Composition through fat mass (FM), lean mass (LM) and soft tissue (ST) analysis. Using this method in our research, we showed that the main reason for the higher body weight of F offspring was the FM increase. Sectional measurements of the masses of the particular depots of the adipose tissue showed that the greatest weight gain in females and males was related to the visceral adipose tissue. 

Similar observations have been published by other authors [22,23,24]. Additionally, both sexes of F offspring had greater LM, as previously was reported by de Albuquerque Maia et al. [24]. Moreover, Madeira et al. [25] documented a positive correlation between LM and BMD, as well as bone microstructure in obese adults with metabolic syndrome, suggesting a relationship between LM and bone health. In line with the assumptions of these authors and regarding the significant increase of LM in F offspring, we considered the impact of LM on bone density. In contemporary literature, the two views of the influence of overweight and obesity on the metabolism of bone tissue clash [7]. It seems that the dominant view is that the adipose tissue harms the skeletal system, increasing bone susceptibility to fracture [5,10,26]. 

However, studies documenting the beneficial effect of weight gain on the metabolism of the skeletal system that result in an increase of BMD and bone mechanical strength cannot be omitted [9,27]. The mentioned DXA analysis allows for a planar evaluation of bone tissue. During the test, we obtain two main values, i.e., BMC, which is the effect of calcium absorption and its deposition in the bone [28], and bone area, mainly determined by the synthesis of proteins responsible for bone structure [19]. The proportion of BMC to bone area allows for estimation of BMD expressed in g/cm2 given by DXA. In the presented research, both t.BMC and t.Ar showed a significant increase in values. Still, only t.BMD tended to be higher. It should be assumed that the lack of significant changes of t.BMD may result from a similar proportion of the increase of t.BMC and t.Ar values of the female and male tibia in groups S and F, which means that despite the increased Ca retention in bones and the effective synthesis of bone matrix proteins increasing the bone surface, t.BMD does not change significantly. Interestingly, a similar relationship was observed by de Albuquerque Maia et al. [24]. 

DXA, despite its advantages, does not allow for the spatial assessment of bone tissue and provides a 2D areal rather than a volumetric calculation of bone size and mass. A three-dimensional analysis is possible using the pQCT and the µCT. The pQCT allows for a separate evaluation of the cortical and the trabecular bone tissue in a volumetric approach in any chosen bone location (Figure 3).

In our research, the evaluation of the cortical bone tissue was carried out in the center of the mid-shaft and significantly higher values of the Ct.vBMD and the Ct.BMC in the female and male offspring of group F were discovered. The results of our research are consistent with previous studies [24]. It is worth noting that the high-energy diet applied to parents had a significant impact on the geometric properties of the tibia of both sexes of offspring, as indicated by the increased thickness of the cortical bone tissue (Ct.Th).

The trabecular bone tissue was also assessed by tomography. It must be underlined that trabecular bone tissue is more sensitive to any treatment, and metabolic processes affecting mineralization or bone structure are more intensive. The analysis in the proximal diaphysis also showed higher values of Tb.vBMD and Tb.BMC in group F female and male offspring. However, the data obtained during the pQCT measurements did not explain the grounds of these changes. This is possible only when the microarchitecture of the trabecular compartment is studied using µCT. 

This showed that after applying a high-energy diet in parents, the bone trabeculae analyzed in the proximal part of the tibia from the F offspring were characterized by a greater thickness (Tb.Th) and a smaller separation between them (Tb.Sp), despite the reduction in their number (Tb.N). Changes in Tb.Th and Tb.Sp in combination with an increase of bone volume fraction (BV/TV), and decrease of bone surface density (BS/TV), as well as structure model index (SMI) indicate that the trabecular compartment is denser in the proximal tibia of females and males from group F [29]. As previously mentioned, numerous studies demonstrated that the increased depot of adipose tissue has a destructive effect on bones, leading to a decrease in their mechanical strength and an increased susceptibility to fracture. This thesis was proven by the observations of obese children at increased risk of fractures [30], who have reported more frequent bone fractures [31,32] and repeated fractures [33]. 

It is believed that increased bone fragility in obese children may result from a limited dietary intake of vitamin D and calcium, as well as a tendency to a sedentary lifestyle [34,35,36]. On the other hand, cross-sectional and longitudinal studies indicate that the relationship between the mass of the adipose tissue and the quality of bone tissue may differ at different stages of growth and development of the organism. It has been suggested that in the early stages of life, the increase in FM and LM has a beneficial effect on bone tissue [37], and the moment of change of the nature of the impact may depend on the exceeding of the hypothetical “fat threshold” at which an overabundance of fat tissue imparts a deleterious effect on the growing skeleton [38,39]. 

This is also confirmed by our study, in which the analysis of mechanical resistance was carried out using pQCT and the three-point bending test. The pQCT test allows for the in vivo analysis of bone tissue, and the obtained Strength-Strain Index (xSSI) shows a high correlation with the strength parameters obtained during loading tests [40]. The bone analysis carried out in our studies on the day of weaning, with the use of pQCT and three-point bending test, confirms the above thesis, as it was demonstrated that the mechanical strength of the tibia shaft of offspring rats from group F was significantly higher. 

The concentration of the CTX-I reflects the intensity of osteoclastic bone tissue resorption. The osteocalcin level, as a bone-specific protein secreted by osteoblast, reflects the metabolic activity of osteoblasts. The activity of the bone-specific alkaline phosphatase (bALP) [41] has a similar diagnostic value. In our research, the abovementioned biochemical markers of bone metabolism confirm that in the female and male offspring of group F, increased body weight has a beneficial effect on the development and mineralization of the bone tissue. 

Vitamin D is well known to play a key role in bone metabolism and calcium homeostasis. It is clear that in adults there is a close correlation between vitamin D level and obesity. Herein, it appears that obesity is associated with a reduction of the 25(OH)D3 concentration [42]. In our study, we found that the level of the 25(OH)D3 is sexually different. The offspring of F females tended to the lower 25(OH)D3 levels, which is consistent with the observations of de Albuquerque Maia [24]. In the male offspring from group F, a 7% increase of the 25(OH)D3 concentration was found. These data confirm the earlier study performed by de Albuquerque Maia et al. [43] carried out on weaned rats. However, the authors did not provide information about the sex of the rats used in the experiment. At this moment, it is difficult to explain the essence of the sexual difference of the 25(OH)D3 concentration, and this problem requires further research.

However, our research has some important limitations. In line with the assumptions, the starting point of our experiment was the simultaneous induction of maternal and paternal obesity. The obtained results reflect the programming influence of obesity of both parents on the skeletal system of their offspring. In the literature, the most frequently described relationships on organogenesis and metabolism of the offspring are that of maternal obesity or of significant changes in nutrition in the period preceding or during pregnancy [44]. Studies of the impact of father obesity on offspring programming effects are not numerous, but they confirm the existence of such a relationship. McPherson et al. [45] suggest that paternal overweight/obesity induces paternal programming of offspring phenotypes likely mediated through genetic and epigenetic changes in spermatozoa. Human and animal studies have shown that paternal obesity increases the risk of diabetes type II and cardiovascular diseases [46]. Therefore, it is important and necessary to extend the research to assess the exclusive impact of fixed paternal obesity on the programming of skeletal changes in offspring.

## 5. Conclusions

It should be emphasized that fixed parental obesity exerts a programming influence on the growth, development and mineralization of the skeletal system of the offspring. Moreover, it significantly intensifies mineralization, and as a consequence, increases bone strength. Our research is in agreement with the assumptions but was limited to a specific age group, i.e., weaning day. However, taking into account numerous reports characterizing the effect of adipose tissue on bone tissue, further work is undoubtedly necessary to determine the time intervals determining the metabolic relationship between adipose tissue and bone in subsequent stages of life. Such studies will allow an understanding of the conditions for determining the “fat threshold”, upon which the process of exceeding the destructive effect of adipose tissue on bone metabolism is initiated.

## Figures and Tables

**Figure 1 animals-11-00424-f001:**
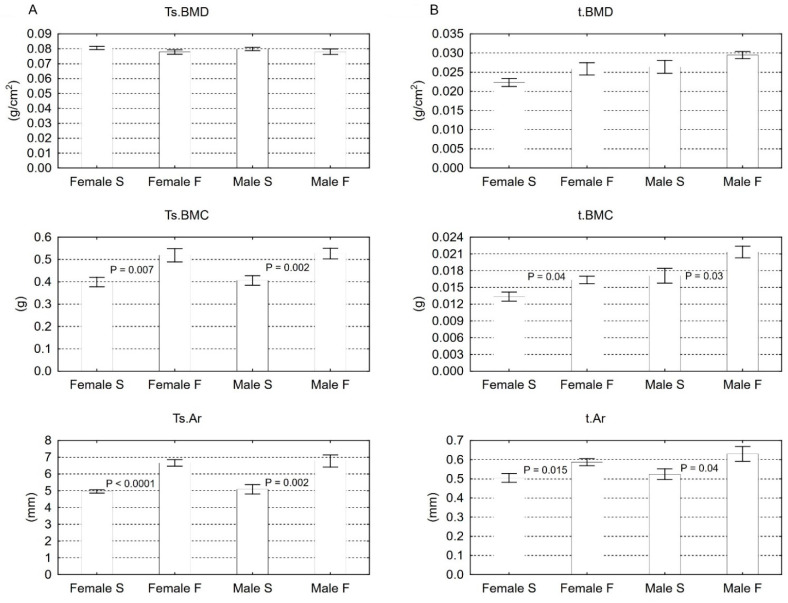
DXA analysis of total skeleton (**A**) and isolated tibia (**B**) of female and male rats on the weaning day. Results are the means ± S.E.M. (*n* = 8). Statistically significant differences between S and F groups were established by the *t*-Student (*p* < 0.05). Abbreviations: Ts.BMD–total skeleton bone mineral density; Ts.BMC–total skeleton bone mineral content; Ts.Ar–total skeleton area; t.BMD–tibia bone mineral density; t.BMC–tibia bone mineral content; t.Ar–tibia bone mineral area.

**Figure 2 animals-11-00424-f002:**
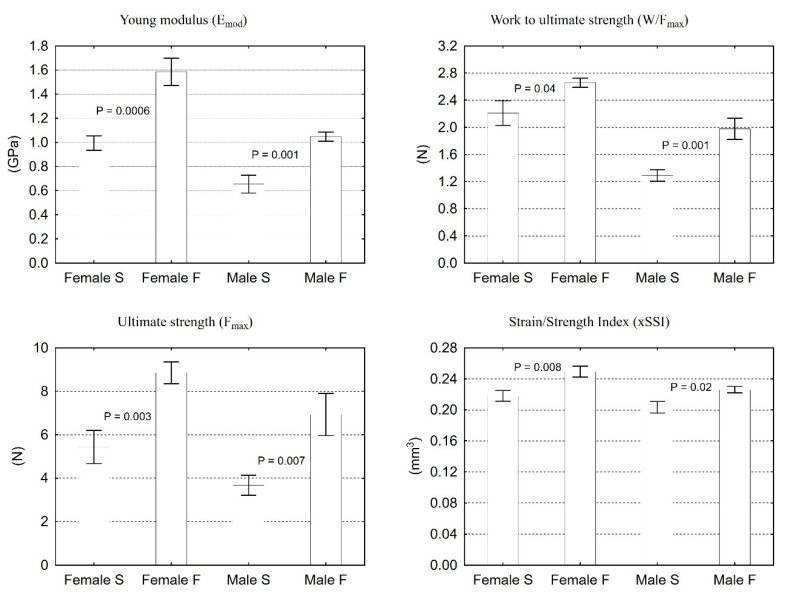
Mechanical parameters of isolated tibia established using three-point bending test and pQCT of female and male rats on the weaning day. Results are the means ± S.E.M. (*n* = 8). Statistically significant differences between *S* and *F* groups were established by the *t*-Student (*p* < 0.05).

**Figure 3 animals-11-00424-f003:**
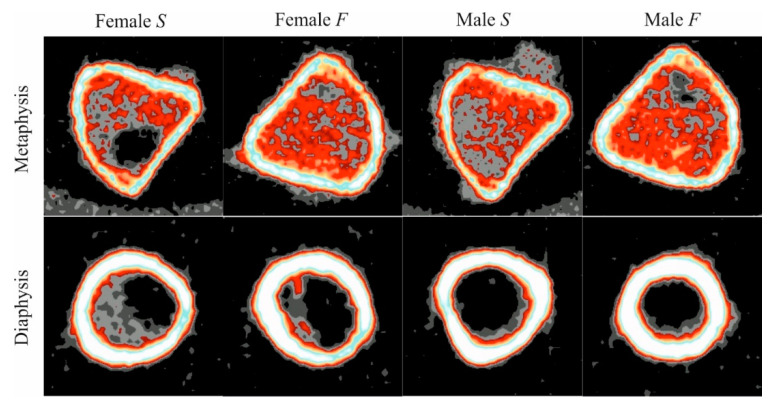
Representative peripheral quantitative computed tomography (pQCT) scan images of tibia performed in the proximal metaphysis and midshaft of the diaphysis of female and male rats on the weaning day.

**Table 1 animals-11-00424-t001:** Standard (*S*) and high-energy diets (*F*) content.

Ingredients	Standard Diet (*S*)	High-Energy Diet (*F*)
Metabolic Energy	11.50 MJ/kg	17.6 MJ/kg
Crude protein	16.00%	20.0%
Crude fat min.	2.80%	21.0%
Crude ash max.	7.00%	5.58%
Crude fiber max.	5.00%	3.86%
Lysine	0.80%	1.2%
Methionine + Cysteine	0.50%	0.76%
Tryptophan	0.190%	0.27%
Calcium	1.10%	1.32%
Phosphorus	0.70%	0.54%
Natrium	0.22%	0.19%
Vitamin A	8000 IU/kg	8000 IU/kg
Vitamin D3	1000 IU/kg	1000 IU/kg
Vitamin E	50 mg/kg	76.9 mg/kg

**Table 2 animals-11-00424-t002:** The body weight (birthday and at weaning), masses of fat tissue depots, weight and length of tibia and parameters of body composition of female and male rats on the weaning day.

Item	Female			Male		
	*S*	*F*	*p*-Value	*S*	*F*	*p*-Value
Body weight (birthday) (g)	8.2 ± 0.2	9.5 ± 0.3	0.004 *	8.5 ± 0.2	9.4 ± 0.2	0.002 *
Body weight (weaning) (g)	60.0 ± 2.3	70.4 ± 1.4	0.002 *	66.0 ± 0.9	85.3 ± 1.1	<0.0001 *
Subcutaneous fat (g)	1.03 ± 0.04	1.25 ± 0.09	0.036 *	1.08 ± 0.08	1.35 ± 0.10	0.046 *
Visceral fat (g)	0.21 ± 0.03	0.54 ± 0.07	0.008 *	0.21 ± 0.03	0.51 ± 0.09	0.003 *
Mesenteric fat (g)	0.38 ± 0.03	0.53 ± 0.03	0.007 *	0.40 ± 0.04	0.51 ± 0.02	0.020 *
Tibia mass (g)	0.104 ± 0.004	0.107 ± 0.003	0.243	0.104 ± 0.003	0.109 ± 0.003	0.187
Tibia length (mm)	23.2 ± 0.4	22.0 ± 0.2	0.301	22.9 ± 0.5	22.2 ± 0.2	0.277
Soft tissue mass (ST) (g)	53.8 ± 2.2	57.9 ± 2.4	0.227	57.5 ± 2.2	61.2 ± 2.2	0.267
Fat tissue mass (FM) (g)	0.060 ± 0.009	0.552 ± 0.079	<0.0001 *	0.072 ± 0.009	1.104 ± 0.085	<0.0001 *
Lean mass (LM) (g)	57.4 ± 1.8	58.4 ± 2.0	0.730	61.4 ± 1.4	63.5 ± 1.7	0.345

Results are the means ± S.E.M. (*n* = 8). Statistically significant differences between *S* and *F* groups were established by the *t*-Student (*p* < 0.05) and marked by *.

**Table 3 animals-11-00424-t003:** The tomographic (pQCT) parameters of the total cross-section of the diaphysis and the proximal metaphysis as well as cortical and trabecular bone tissue of the tibia of female and male rats on the weaning day.

Item	Female			Male		
	*S*	*F*	*p*-Value	*S*	*F*	*p*-Value
pQCT analysis of tibia diaphysis (50% of bone length)						
Tot.BMC (mg/mm)	0.86 ± 0.05	0.92 ± 0.06	0.523	0.86 ± 0.04	0.98 ± 0.03	0.053
Tot.vBMD (mg/mm^3^)	406 ± 13	422 ± 20	0.719	396 ± 18	423 ± 14	0.260
Tot.Ar (mm^2^)	2.15 ± 0.03	2.19 ± 0.08	0.734	2.20 ± 0.10	2.32 ± 0.08	0.378
Peri.C (mm)	5.2 ± 0.1	5.2 ± 0.1	0.749	5.3 ± 0.1	5.4 ± 0.1	0.366
Endo.C (mm)	3.8 ± 0.1	3.7 ± 0.1	0.890	3.8 ± 0.1	3.9 ± 0.1	0.444
Ct.BMC (mg/mm)	0.85 ± 0.03	0.93 ± 0.04	0.032 *	0.86 ± 0.02	0.96 ± 0.02	0.006 *
(Ct.vBMD (mg/mm^3^)	786 ± 7	841 ± 9	0.032 *	814 ± 6	871 ± 9	0.016 *
Ct.Ar (mm2)	1.02 ± 0.02	1.08 ± 0.03	0.133	1.06 ± 0.03	1.11 ± 0.02	0.250
Ct.Th (mm)	0.229 ± 0.002	0.242 ± 0.004	0.036 *	0.234 ± 0.002	0.254 ± 0.003	0.001 *
pQCT analysis of tibia proximal metaphysis						
Tot.BMC (mg/mm)	1.6 ± 0.1	1.9 ± 0.1	0.017 *	1.7 ± 0.1	2.1 ± 0.1	0.006 *
Tot.vBMD (mg/mm^3^)	271 ± 8	322 ± 14	0.007 *	254 ± 9	322 ± 11	0.0003 *
Tot.Ar (mm^2^)	6.0 ± 0.2	6.0 ± 0.3	0.892	6.6 ± 0.4	6.5 ± 0.3	0.789
Tb.BMC (mg/mm)	0.28 ± 0.02	0.40 ± 0.03	0.005 *	0.29 ± 0.03	0.42 ± 0.03	0.012 *
(Tb.vBMD (mg/mm^3^)	92 ± 12	143 ± 14	0.018 *	80 ± 16	140 ± 14	0.015 *
Tb.Ar (mm2)	2.7 ± 0.1	2.7 ± 0.2	0.848	3.0 ± 0.2	2.9 ± 0.1	0.820

Results are the means ± S.E.M. (*n* = 8). Statistically significant differences between S and F groups were established by the *t*-Student (*P* < 0.05) and marked by *. Abbreviations for cortical bone tissue–Tot.BMC-Total slice bone mineral content, Tot.vBMD-total slice volumetric bone mineral density, Tot.Ar-total slice bone area, Ct.Ar-cortical bone area, Ct.BMC cortical bone mineral content, Ct.vBMD-cortical volumetric bone mineral density, Ct.Th-cortical thickness, Peri.C periosteal circumference, Endo.C endosteal circumference. Abbreviations for trabecular bone tissue–Tot.BMC-total slice bone mineral content, Tot.vBMD-total slice volumetric bone mineral density, Tot.Ar-total slice bone area, Tb.Ar-trabecular bone area, Tb.BMC-trabecular bone mineral content and Tb.vBMD-trabecular volumetric bone mineral density.

**Table 4 animals-11-00424-t004:** Micro-CT measurement of the trabecular bone tissue in the proximal tibia metaphysis of female and male rats on the weaning day.

Item	Female			Male		
	*S*	*F*	*p*-Value	*S*	*F*	*p*-Value
TV (mm^3^)	14.8 ± 0.8	15.5 ± 0.1	0.35	15.0 ± 0.5	15.6 ± 0.4	0.43
BV (mm^3^)	8.8 ± 0.3	11.6 ± 0.1	<0.0001 *	8.1 ± 0.2	12.3 ± 0.3	<0.0001 *
BS (mm^2^)	372 ± 16	351 ± 6	0.280 *	398 ± 9	369 ± 5	0.015 *
BS/TV (1/mm)	25.3 ± 0.6	22.6 ± 0.4	0.004 *	26.6 ± 0.5	23.7 ± 0.3	0.0006 *
BV/TV (%)	59.2 ± 1.6	74.2 ± 0.9	<0.0001 *	58.1 ± 0.9	79.5 ± 2.6	<0.0001 *
Tb.Th (mm)	0.21 ± 0.02	0.30 ± 0.02	0.005 *	0.22 ± 0.01	0.32 ± 0.01	<0.0007 *
Tb.N (1/mm)	3.0 ± 0.1	2.4 ± 0.1	0.003 *	2.9 ± 0.1	2.4 ± 0.1	0.0007 *
Tb.Sp (mm)	0.17 ± 0.01	0.13 ± 0.01	0.006 *	0.17 ± 0.01	0.12 ± 0.01	0.014 *
SMI	−1.7 ± 0.1	−3.0 ± 0.1	<0.0001 *	−1.6 ± 0.1	−3.0 ± 0.1	<0.0001 *

Results are the means ± S.E.M. (*n* = 8). Statistically significant differences between S and F groups were established by the *t*-Student (*p* < 0.05) and marked by *. Abbreviations: TV-total volume, BV-bone volume, BS–bone surface, BS/TV-object surface density, BV/TV-bone volume density, Tb.Th-trabecular thickness, Tb.Sp-trabecular separation, Tb.N-trabecular number, SMI-structure model index.

**Table 5 animals-11-00424-t005:** Biochemical markers of bone metabolism and concentration of ionized calcium and phosphorus in blood serum of female and male rats on the weaning day.

Item	Female			Male		
	*S*	*F*	*p*-Value	*S*	*F*	*p*-Value
Osteocalcin (ng/mL)	10.5 ± 0.1	11.2 ± 0.1	0.001 *	10.1 ± 0.2	10.9 ± 0.1	0.005 *
bALP U/L	517 ± 43	803 ± 59	0.002 *	539 ± 27	670 ± 53	0.002 *
CTX-I (ng/mL)	30.9 ± 1.2	26.6 ± 0.4	0.003 *	28.0 ± 0.5	23.0 ± 0.9	<0.0001 *
25 OH D3 (ng/mL)	49.0 ± 2.6	45.9 ± 2.9	0.424	44.3 ± 1.8	47.5 ± 1.9	0.042 *
Phosphorus (mmol/L)	11.7 ± 0.3	12.6 ± 0.8	0.363	12.6 ± 0.6	12.2 ± 0.5	0.329
Calcium (mmol/L)	11.5 ± 0.1	11.6 ± 0.2	0.750	11.8 ± 0.3	11.5 ± 0.1	0.942

Results are the means ± S.E.M. (*n* = 8). Statistically significant differences between *S* and *F* groups were established by the *t*-Student (*p* < 0.05) and marked by *. Abbreviations: bALP–bone-specific alkaline phosphatase, CTX-I-C-terminal telopeptides of type I collagen.

## Data Availability

The data presented in this study are available on request from the corresponding author.

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
