# Peer review of "Programming Effect of the Parental Obesity on the Skeletal System of Offspring at Weaning Day"

_animals, 2021, doi:10.3390/ani11020424_

Round 1
Reviewer 1 Report
Minor comments to authors.
Line 32: mechanical strength was also analyzed.
Line 77: I suggest to rephrase the objective of the study so it reads better: The study focused on determining whether and to what extent the established dietary-induced obesity in parents before fertilization exerts a programming effect on the development of the female and male offspring's skeletal system on the weaning day.
Line 88: energy values
Line 92: provide reference for the time necessary to induce obesity. Also there is no data/comment indicating tat the diets worked.
Line 103: provide storage conditions
Line 116: structures
Line 313: Please comment on the age of the experimental units. Looks like these studies were in performed on adults.
Author Response
The authors would like to express their gratitude to the reviewer for the time devoted and thorough evaluation of the work. The authors are very grateful for all suggestions and comments.
With best regards
Radoslaw P. Radzki
The comments to the revisions:
Line 32 Abstract has been completed with information concerning strength tests.
Line 77 The authors thank you very much for the suggestion to re-edit the objective of the study. The introduced changes are identical to those suggested by the reviewer.
Line 88 The phrase "energy value" has been replaced with "energy values"
Line 92 The materials and methods section has been supplemented with two citations of the literature, which confirm the methodical approach allowing dietary-induced obesity. The values of the body weight changes in females and males of the parental generation have been supplemented in the text of the materials and methods section.
Longhi, R .; Almeida, R.F .; Machado, L .; Duarte, M .; Souza, D.G .; Machado, P .; de Assis, A.M .; Quincozes-Santos, A .; Souza, D.O., Effect of a trans fatty acid-enriched diet on biochemical and inflammatory parameters in Wistar rats. Eur J Nutr 2017, 56, 1003-1016 10.1007 / s00394-015-1148-y.
Lac, G .; Cavalie, H .; Ebal, E .; Michaux, O., Effects of a high fat diet on bone of growing rats. Correlations between visceral fat, adiponectin and bone mass density. Lipids Health Dis 2008, 7, 16 10.1186 / 1476-511X-7-16.
Line 103 The Information on the storage conditions of biological material has been revised. "After sacrifice, the isolated tibia was frozen (-30 ° C) for further analysis."
Line 116 The phrase "structure" has been replaced with "structures"
Line 313 We are sorry but the authors do not understand this comment. Probably the wrong number of text line was entered.
Reviewer 2 Report
Article entitled “Programming effect of the parental obesity on the skeletal system of offspring at weaning day” by Radski_Wolska et al. brings new knowledge in the area of parental obesity effects and high fat diet treatments to the weaning stage on the skeletal system of progeny. I think that the results and the scope of this study is within the remit of ANIMALS journal. I would suggest a minor revision as follows:
Simple summary:
I would avoid sentences with vague terms such as »more intense« and higher mechanical strength«. Instead indicate what results from your study were statistically significant and contributed most new knowledge.
Line 38: “Biomechanical” analyses of serum – probably a mistake, you meant “Biochemical” analyses?
Line 40 – spell out abbreviated symbols such as DXA abd BMC
Line 58 adipose tissue..«largest« endocrine organ – I believe skeletal muscle is the largest endocrine organ with all secreted myokines etc.not the adipose tissue
Line 84: Animal subline:
Please indicate in the Material and methods the subline of the Wistar rats – e.g. if it was bought from the Charles rivers laboratory its code is Crl:WI, if it is from Harlan-Envigo the subline name is Hsd:WI and so on. If it was bought in the past from some vendor and is now bred in your facility indicate that as well. As the line is outbred it may differ genetically between sublines and hence affect reproducibility of results, so I suggest to be as explicit as possible.
Line 88 – »live« diet – what is meant by »live« not a very common term for a diet?
Line 182 Perhaps add at the end of Table 2 title …«of offspring at weaning« , just to make it clearer – I'd suggest to do similar in other results sections, so that figures and tables are more self-explanatory
Line 164 - Gene, protein nomenclature:
Osteocalcin – is a synonym. Please define when first intoduced also the official name of the gene or protein. Osteocalcin is also a synonym for two genes in mice it may as well be the same in rats - Bglap2 and Bglap3 (official name: bone gamma-carboxyglutamate protein 2 or 3). Define which of the two (or both?) were measured in your study by the assay used.
Discussion
I would encourage authors to include a short paragraph or so in discussing limitations and pitfalls of the current study. Namely, the design of this study captured various factors in the term »parental obesity programming« - it involved probably mainly maternal prenatal obesity and postnatal changes in milk composition effects on the weaned offspring bone characteristics. But also, paternal obesity effects can not be ruled out. Moreover, pups were starting to eat full fat diet before weaning and this caloric input too could have an effect. These issues and cautionary notes in interpretation of the results would be approapirate as well as discussion how to disentangle these effects in future experimental designs.
Author Response
The authors would like to express their gratitude to the reviewer for the time devoted and thorough evaluation of the work. The authors are very grateful for all suggestions and comments.
With best regards
Radoslaw P. Radzki
The comments to the revisions:
The phrase "more intense and higher mechanical strength" in the context of the isolated tibia was used in a simple summary. As recommended for authors, this part of the manuscript "should be written for a lay audience, i.e., no technical terms without explanations". The phrases bone mineralization and mechanical strength should be generally understood. However “mechanical strength “ was changed for “fracture resistance”.
Line 38 Thank you for pointing out this error. The word "Biomechanical" has been changed to "Biochemical"
Line 40 Thank you, this is a very important suggestion. The BMC symbol was explained and the DXA explanation was on line 37
Line 58 Accepting the reviewer's suggestion, we propose introducing an alternative formulation. Instead of "but is also the largest endocrine organ," we suggest "but is also one of the largest endocrine organs,".
Line 84 The information about animals were supplied by the sentence (outbred herd in Center of Experimental Medical in Bialystok Medical University - Cmdb: Wi, originated from Charles River)
Line 88 The word "live" is redundant here and has been removed as unnecessary. Unfortunately, its presence was confirmed after sending the manuscript, for which the authors apologize and thank the reviewer for this remark.
Line 182 As suggested by the reviewer, the titles of tables and figures have been completed. After making the changes, they are presented as follows:
Table 2
Previously: The body weight (birthday and at weaning), masses of fat tissue depots, weight and length of tibia and parameters of body composition.
Recently: The body weight (birthday and at weaning), masses of fat tissue depots, weight and length of tibia and parameters of body composition of females and males rats on the weaning day
Figure 1
Previously: (first sentence) DXA analysis of total skeleton (panel A) and isolated tibia (panel B) densitometry.
Recently: DXA analysis of total skeleton (panel A) and isolated tibia (panel B) of females and males rats on the weaning day
Table 3
Previously: The tomographic (pQCT) parameters of the total cross-section of the diaphysis and the proximal metaphysis as well as cortical and trabecular bone tissue of the tibia.
Recently: The tomographic (pQCT) parameters of the total cross-section of the diaphysis and the proximal metaphysis as well as cortical and trabecular bone tissue of the tibia of females and males rats on the weaning day.
Table 4.
Previously: Micro-CT measurement of the trabecular bone tissue in proximal tibia metaphysis Recently: Micro-CT measurement of the trabecular bone tissue in proximal tibia metaphysis of females and males rats on the weaning day.
Figure 2.
Previously: Mechanical parameters of isolated tibia established using three-point bending test and pQCT.
Recently: Mechanical parameters of isolated tibia established using three-point bending test and pQCT of females and males rats on the weaning day.
Table 5.
Previously: Biochemical markers of bone metabolism and concentration of ionized calcium and phosphorus.
Recently: Biochemical markers of bone metabolism and concentration of ionized calcium and phosphorus in blood serum of females and males rats on the weaning day.
Figure 3.
Previously: Representative peripheral quantitative computed tomography (pQCT) scan images of tibia performed in the proximal metaphysis and midshaft of the diaphysis.
Recently: Representative peripheral quantitative computed tomography (pQCT) scan images of tibia performed in the proximal metaphysis and midshaft of the diaphysis of females and males rats on the weaning day.
Line 164 Description for osteocalcin (OC) has been completed. However, in bone metabolism studies, osteocalcin level is used as a marker of bone formation. OC is specifically expressed in osteoblasts and is the most abundant non-collagenous protein in bone. OC presents a high affinity to Ca2 + by carboxylation of three glutamic acids. In studies regarding the metabolism of bone tissue, the locution "osteocalcin analysis" does not require further explanation.
In mice, three osteocalcin genes have been established. These include Bglap, Bglap2 (which are indeed synonymous with OC) and are present in osteoblasts, while Bglap3 is not present in bone tissue. However, Bglap3 is found in kidneys, lungs and testes. Interestingly, only one osteocalcin gene (BGLAP) has been identified in rats and humans.
As expected by the reviewer, the discussion was supplemented with a fragment, which indicated the limitations of the presented research. Below is a text of this part of the discussion and added references.
“However, our research has some important limitations. In line with the assumptions, the starting point of our experiment was the simultaneous induction of maternal and paternal obesity. The obtained results reflect the programming influence of obesity of both parents on the skeletal system of their offspring. In the literature, the most frequently described relationships determining maternal obesity or significant changes in nutrition in the period preceding or during pregnancy on organogenesis and metabolism of the offspring (1). Researches aiming the programming effects of paternal obesity on offspring are scarce, but it does prove such the relationship. McPherson et al. (2) suggest that paternal overweight/obesity induces paternal programming of offspring phenotypes likely mediated through genetic and epigenetic changes in spermatozoa. Human and animal studies have shown that paternal obesity increases the risk of diabetes type II and cardiovascular dieseases (3). Therefore, it is important and necessary to extend the research to assess the exclusive impact of fixed paternal obesity on the programming of skeletal changes in offspring.”
- Cirulli F, Musillo C, Berry A. Maternal Obesity as a Risk Factor for Brain Development and Mental Health in the Offspring. Neuroscience. 2020; 447: 122-35. 2. McPherson NO,
- Fullston T, Aitken RJ, Lane M. Paternal obesity, interventions, and mechanistic pathways to impaired health in offspring. Ann Nutr Metab. 2014; 64 (3-4): 231-8. 3.
- Eberle C, Kirchner MF, Herden R, Stichling S. Paternal metabolic and cardiovascular programming of their offspring: A systematic scoping review. PLoS One. 2020; 15 (12): e0244826.
As suggested by the reviewer, the content of the "Conclusion" section has also been changed